# Quantitation of Water Addition in Octopus Using Time Domain Reflectometry (TDR): Development of a Rapid and Non-Destructive Food Analysis Method

**DOI:** 10.3390/foods11060791

**Published:** 2022-03-09

**Authors:** Bárbara Teixeira, Helena Vieira, Sandra Martins, Rogério Mendes

**Affiliations:** 1Department for the Sea and Marine Resources, Portuguese Institute for the Sea and Atmosphere, Avenida Doutor Alfredo Magalhães Ramalho 6, 1495-165 Algés, Portugal; helenavieira27@gmail.com (H.V.); sandra.regalado@ipma.pt (S.M.); rogerio@ipma.pt (R.M.); 2Interdisciplinary Center of Marine and Environmental Research (CIIMAR), University of Porto, Rua das Bragas 289, 4050-123 Porto, Portugal; 3Research Unit of Organic Chemistry, Natural and Agro-Food Products, Aveiro University, Campus Universitário de Santiago, 3810-193 Aveiro, Portugal

**Keywords:** rapid methods, dielectric properties, conformity, cephalopods, adulteration, quality control

## Abstract

A rapid and non-destructive method based in time domain reflectometry analysis (TDR), which detects and quantifies the water content in the muscle, was developed for the control of abusive water addition to octopus. Common octopus samples were immersed in freshwater for different periods (0.5–32 h) to give a wide range of moisture contents, representing different commercial conditions. Control and water-added octopus were analyzed with a TDR sensor, and data correlated with moisture content were used for calibration and method validation. A maximum limit of moisture content of 85.2 g/100 g in octopus is proposed for conformity assessment, unless the label indicates that water (>5%) was added. Calibration results showed that TDR analysis can discriminate control and water-added octopus, especially for octopus immersed for longer periods (32 h). In addition, moisture content can be quantified in octopus using only TDR analysis (between 80 and 90 g/100 g; RMSE = 1.1%). TDR data and correlation with moisture content show that this non-destructive methodology can be used by the industry and quality control inspections for assessment of octopus quality and to verify compliance with legislation, promoting fair trade practices, and further contributing to a sustainable use of resources.

## 1. Introduction

In view of its worldwide presence in tropical and temperate seas, the common octopus (*Octopus vulgaris*) is the most significant commercially harvested octopus species, usually traded as fresh, dried, salted, or mostly frozen [1]. Octopus is highly valued in Mediterranean, South American and East Asian countries, and the species supports artisanal as well as industrial fisheries. Octopus is typically valued for its sensory properties, freshness and intrinsic quality. In recent years, the increasing consumer demand for this species has caused prices to rise throughout the distribution chain, which has led octopus to be one of the most important fishery resources in terms of value in Southern European countries [2].

Seafood is often the target of practices that may affect product integrity, especially in those species with high added value [3,4]. One example is the abusive and non-reported water addition to allegedly compensate for moisture losses [3,5]. Due to its high price and intrinsic physiological characteristics, octopus is prone to mislabeling and abusive industrial processing, both by fishermen and seafood processors. Since octopus is considerably hyperosmotic compared with the seawater in which it lives, and osmotic uptake over the general body surface is possible [6], seafood processors soak octopus in water or in solutions with additives to increase the yield.

European Union labeling rules, which ratified the mandatory Quantitative Ingredients Declaration [7], enable citizens to obtain comprehensive information on the content and composition of food products, which can help consumers to make an informed choice while purchasing their foodstuffs. In the case of seafood, an indication of the presence of added water, which makes up more than 5% of the weight of the finished product, must be included in the label of the food in the cases of fishery products and prepared fishery products, which have the appearance of a cut, joint, slice, portion, fillet or of a whole fishery product [7].

Consumers complaints for an enormous weight loss of octopus after cooking lead the media to raise suspicions regarding excessive water addition that could be deemed as adulteration and result in economic fraud for the buyer [8,9]. Likewise, market studies regarding the addition of water have confirmed these abusive practices [10,11]. A comparative test held in Portugal with 25 commercial samples of prepackaged deep-frozen octopus (*Octopus vulgaris*) showed that most of the products (92%) presented significantly higher moisture and lower protein contents than unprocessed samples, signaling water soaking processing [11]. In another study with octopus, Mendes et al. [12] also reported artificial weight increases of up to 50.8% and cooking losses, in general, 20% higher than in unprocessed samples. Consumers do not expect a considerable amount of water in these foods (more than the one stated in the label), for which rapid and accurate quality control methods are necessary for the detection of fraudulent octopus processing and buyer’s expectations protection.

Destructive analytical methods for the quantitation of moisture and protein contents are time consuming and need to be supported by baseline levels for an adequate control of water addition in commercial octopus products [12]. However, for a prompt evaluation of abusive practices before first sale, as early as the primary processing stage for product qualification in the industry, or in fresh products in markets, rapid and non-destructive methods would be desirable.

The measurement of moisture, water activity, ripeness/spoilage, physico-chemical composition and state, and detection of adulterants, using microwave dielectric spectroscopy, has been proposed in several food products [13,14,15]. Microwave dielectric spectroscopy, using specifically time domain reflectometry (TDR), is a methodology that probes the dielectric properties of the material under test in a broad microwave frequency range (100 MHz–10 GHz), by measuring the voltage signal reflected by the sample as a function of time, followed by multivariate analysis of the spectra obtained [16]. As water is usually the most abundant dipolar molecule in food products, its interaction with oscillating electric fields in the range of microwaves affects the vibrational dynamics of these molecules and influences and controls the microwave dielectric spectrum [17]. Therefore, differences in the amount of water or aggregation state in food are therefore likely to change the dielectric spectrum at a given frequency range and can be detected by this technology. In addition, the dielectric method for assessment of meat has several advantages: it is non-destructive, easy, rapid, effective, reliable and practical [18].

Using TDR technology, Mendes et al. [19] proposed a detection method for artificially water-added *Octopus vulgaris* and concluded that simultaneous measurement of electrical conductivity and dielectric properties could be used to distinguish water-added octopus among processed octopuses. Conversely, Lee et al. [20] also found that the measurement of the dielectric properties of *Octopus minor* using the coaxial probe method has a great possibility to distinguish between normal frozen octopuses and artificially water-injected frozen octopuses. However, as far as can be determined from the published literature, there was not an attempt to determine quantitatively how much water was artificially added into octopus by measuring the microwave dielectric spectra properties with time domain reflectometry.

Considering the significance of octopus quality, the promotion of fair trade and the protection of consumer trust, the main objective of this study was to develop a new rapid and non-destructive method, involving time domain reflectometry measurements, that can be used in quality control not only for detection, but specifically for quantitation of the water content in water-added octopus.

## 2. Materials and Methods

### 2.1. Raw Material, Processing, and Sampling

A schematic representation of water addition trials with octopus is shown in Figure 1. Common octopus *Octopus vulgaris* was captured in Peniche (Portugal west coast) during 2019–2021, and the weight of individuals was 1.2 ± 0.3 kg. A total of seven trials were performed, with octopus and the number of specimens used in each trial varied between 12 and 14. Six trials (n = 77) were used for the calibration of the time domain reflectometry method, and one trial (n = 12) was performed for the validation.

Water addition trials followed the previous experience of Mendes et al. [12]. To suppress individual variability during water addition studies, each octopus was divided into four equal parts, and each comprised two arms and a portion of the head. One part was always used as a control, while the correspondent remaining three parts were submitted to different processing time conditions (e.g., trial 1: 1, 4, and 16 h; Figure 1). Octopus samples were immersed in freshwater (octopus:water 1:2 w:v) for different periods (0.5, 1, 2, 4, 16, 24, or 32 h) under refrigeration (3 ± 1 °C). The various treatments were intended to give a wide range of moisture contents in octopus samples. TDR analysis was performed in all octopus samples (control and water-added) with RFQ-Scan® equipment, and then samples were vacuum packed and frozen (−20 °C) until further analysis (moisture, protein, electrical conductivity).

### 2.2. Weight Changes, Moisture and Protein Contents

Weight changes were determined as percentage weight differences of octopus samples before and after water addition trials. Moisture content was determined by drying the minced octopus sample (10 g) overnight at 105 °C, using the standard gravimetric analysis [21]. Crude protein content was determined in minced octopus sample (250 mg) by the Dumas combustion method according to Saint-Denis and Goupy [22] in a LECO FP-528 protein/nitrogen analyzer (LECO Corp., St Joseph, MI, USA) calibrated with ethylene diamine tetra-acetic acid (nitrogen: 9.57 ± 0.03 g/100 g). A conversion factor of 6.25 was used. All determinations were performed in duplicate.

### 2.3. Electrical Conductivity

Owing to the non-availability of dedicated surface conductivity probes specially designed for non-destructive octopus analysis, alternative methods adapted to be as close as possible to the objective intended were used. Minced octopus muscle (10 g) was homogenized with 10 mL of Merck-Millipore Milli-Q water (Darmstadt, Germany) in a Polytron 5000 blender (40 s, 15 000 rpm), and then electrical conductivity was measured in the homogenate. An Orion 162 conductivity meter equipped with an Orion 018010 two-electrode electrical conductivity cell (Orion Research Inc., Boston, MA, USA) was used for electrical conductivity determinations. All measurements were made in duplicate.

### 2.4. Time Domain Reflectometry

#### 2.4.1. RFQ-Scan^®^ Analysis

TDR measurements were performed with a patented smart system based on the principle of dielectric spectroscopy, namely, the RFQ-Scan^®^ (Radio Frequency Quality Scan) developed by Sequid GmbH (Bremen, Germany). The device generates a step-like voltage signal of 2.56 ns (total duration) with a rise time of approximately 100 ps. The signal propagates through a coaxial sensor with an open-ended termination, where it interacts with the sample. The reflected signal in the time domain is recorded and evaluated using multivariate statistical methods. In the frequency domain, the signal is characterized by a broad bandwidth of approximately 5 GHz. The time domain spectrometer has been described in more detail by Schimmer and Knöchel [23]. The samples were kept under refrigeration (3 ± 1 °C) during sample preparation and before the measurements, and thus the temperature was considered nearly constant, and its influence can be neglected. Eight scans were made in distinct areas on the skin of octopus in order to perform an appropriate averaging for each sample.

#### 2.4.2. Calibration and Validation of TDR Analysis

Two approaches were followed to calibrate the TDR analysis for (i) detection of water-added octopus, and (ii) to estimate the moisture content of this species. A total of six trials (n = 77 for control and n = 231 for water-added samples) were used for both calibrations. Water-added samples consisted of octopus immersed in freshwater for different periods to achieve different moisture contents in octopus samples. After TDR analysis, only a subset in the most relevant part of the reflected TDR signals was selected, between 1.01 and 1.33 ns (nine points).

For the detection of water-added octopus, i.e., to discriminate between control and water-added octopus, principal components analysis was performed. The first two principal components were plotted against each other, and two clusters were obtained heuristically with an ellipsoid shape, with parallel long axis. To determine the cluster affiliation of a new sample, the maximum likelihood of being classified as water added was determined from the actual distance to the separation line equidistant to the two clusters. A sample with a maximum likelihood value close to 0 belongs to the control cluster, while close to 1 belongs to the water-added cluster, and values were expressed as percentage.

Regarding the calibration for the estimation of moisture content of octopus, a multiple linear regression was performed. Principal components were used, instead of TDR variables, to eliminate the collinearity among TDR variables. The moisture content determined by the destructive method was used, as well as the scores of the first two principal components obtained from the TDR data.

For validation of performed calibrations, a new trial with *O. vulgaris* was conducted. In this validation trial, octopus samples (n = 12 for control and n = 36 for water-added octopus) were analyzed with the RFQ-Scan® equipment, and for an unbiased analysis, the operator did not had access to the results of any calibration. Then, the data were processed similarly and plotted in the principal components plot with the clusters, which was dependent on TDR data acquired, and the maximum likelihood values of being classified as processed were determined. In addition, the moisture content was estimated taking into account the multiple linear regression obtained in the second calibration, and the results were compared with the moisture content determined by the destructive method.

### 2.5. Statistical Analysis

Differences in moisture and protein contents, M/P ratio, and electrical conductivity among control and water-added octopus samples (different processing time conditions) were evaluated by means of a general linear model (one-way analysis of variance, ANOVA). Multiple comparisons were performed by using Tukey’s honest significant difference test. Moreover, the Pearson correlation coefficient test was performed to test several variables (e.g., water uptake vs. moisture; principal components scores vs. moisture) for correlation. Statistical analyses were performed at a 0.05 level of probability using STATISTICA Version 10 (StatSoft, Inc., Tulsa, OK, USA).

A principal component analysis (PCA) was performed to evaluate TDR data acquired with the RFQ-Scan^®^ device. Following removal of outliers from the set of eight scans per sample, measurements were averaged and subjected to PCA as described by Schimmer et al. [24]. TDR data processing was performed using MATLAB Version 7.6 (The Math-Works, Inc., Natick, MA, USA).

## 3. Results and Discussion

### 3.1. Characterization of Control Octopus Samples

Moisture and protein contents in the 77 control octopus samples showed a variation range between 79.7 and 85.8 g/100 g and 11.0 and 17.0 g/100 g, respectively. Moisture and protein data followed a normal distribution (Figure 2) and a mean ± SD moisture and protein contents of 82.2 ± 1.3 and 14.6 ± 1.2 g/100 g were determined, respectively. A mean M/P ratio value of 5.7 ± 0.6 (range 4.7–7.8) was also obtained. Moisture and protein contents of test samples were within the data variation range reported by Mendes et al. [12] in 265 unprocessed fresh octopus samples from fishing grounds on the Portuguese coast (moisture 76.2–85.4 g/100 g; protein 12.0–19.1 g/100 g) and were therefore shown to be representative of control octopus products. Marginal differences in the M/P ratio (3.9–6.9) may be justified by the fact that, in this study, frozen octopus samples (thawed before experiments) were also included, and possibly water was lost during thawing.

Taking into account the conformity assessment of octopus samples from the Portuguese fishing grounds, moisture, protein, and M/P data from control samples of this study were aggregated with the baseline data from Mendes et al. [12]. Mean ± SD moisture and protein contents and an M/P ratio of 81.0 ± 1.7 g/100 g, 16.2 ± 1.7 g/100 g and 5.1 ± 0.6, respectively, were obtained (n = 342). Confidence intervals (95%) for octopus moisture and protein contents were 77.6–84.4 and 12.8–19.6 g/100 g, respectively. Specifically in relation to added water, European Union Regulation (EU) no. 1169/2011 (on the provision of food information to consumers) considers in Annex VI (mandatory particulars accompanying the name of the food) that, in the case of fishery products and prepared fishery products, which have the appearance of a cut, joint, slice, portion, filet or of whole fishery products, the name of the food shall include an indication of the presence of added water if the added water makes up more than 5% of the weight of the finished product. The limit level of moisture in octopus for conformity assessment of commercial products was determined as 85.2 g/100 g, based on the highest value of the 95% confidence interval (84.4 g/100 g) plus 5% added water.

Different seasons [25], origins [26], and feeding habits [27] were shown to vary the proximate composition of cephalopod species, including *O. vulgaris*. Although several factors might affect the proximate composition, the range of values obtained in the current study were comparable to results published previously [25,26,27]. Differences in feeding habits throughout seasons, and different qualities as a result of differentiated conservation before analysis (fresh vs. frozen octopus) might explain the variation in the moisture and protein values obtained in the control octopus.

### 3.2. Water Uptake and Moisture Content in Water-Added Octopus

Water addition trials for preparation of octopus samples with a wide range of moisture contents showed that water uptake (weight gain) induced significantly different weight gains, which increased with the increase in immersion time (data not shown). Almost all octopus samples immersed for 2 h or longer periods (126 in 130 octopus samples) had a water uptake of at least of 5%, and reached the limit after which the presence of added water must be declared in the product label [7]. The weight gain after 24 and 32 h water addition processing was 29.7 ± 5.5% and 31.7 ± 8.5%, respectively. Apart from this, significant induced weight gain water uptake also has a huge impact in the final product after cooking, resulting in a further shrinkage of octopus [12], thus deceiving consumers as they buy water for the price of octopus.

Overall, different processing immersion times allow for one to obtain octopus samples with moisture values covering a broad range of values. Moisture contents in water-added octopus ranged from 82.5 to 90.6 g/100 g (Table 1). A previous study about water uptake with octopus showed that average moisture contents ranged from 80.5 to 82.7 g/100 g in control samples, and comparable moisture contents were obtained after 1, 4, and 16 h of immersion in freshwater [12]. Moreover, the current study shows that moisture content increases even further, after 16 h of immersion.

Although moisture contents were significantly different between control and water-added octopus (Table 1), there is a range of moisture values (82–86 g/100 g) shared by both. In general, most water-added octopus processed for 0.5–4 h fitted in this group of moisture values between 82–86 g/100 g, and thus cannot be discriminated from control ones based only on the evaluation of moisture content. Moreover, the addition of ca. 15% of water could still result in octopus with moisture contents within the range of values of control octopus.

Using the traditional destructive method of moisture determination, water uptake can be estimated as a function of moisture content, as a significant direct correlation was found (water uptake = 4.149 × moisture content (g/100 g) − 345.05; r = 0.72; *p* value < 0.01).

### 3.3. Protein and Moisture/Protein Ratio in Water-Added Octopus

Protein and moisture/protein ratio of control and water-added octopus samples are presented in Table 1. As expected, protein content of octopus decreased gradually as immersion time increased, from 14.6 ± 1.2 g/100 g in control samples to 9.1 ± 0.7 g/100 g in octopus immersed for 32 h. Considering the range of protein values in control octopus, most samples immersed in freshwater for 0.5–16 h cannot be distinguished from control ones, as protein contents are within the natural variation of values.

Regarding the moisture/protein ratio of water-added octopus, this ratio increased with the increase in immersion time. A moisture/protein ratio of 9.8 ± 0.9 was determined for the 32 h treatment. Similarly, the use of this ratio per se cannot be used to distinguish control from water-added octopus if short-time (0.5–4 h) treatments are applied, due to the range of values determined in control octopus.

Water uptake can be predicted as a function of moisture/protein ratio, as a significant positive correlation was found (water uptake = 5.4086 × M/P ratio − 27.698; r = 0.67; *p* value < 0.01). A previous study showed a lower dispersion (r = 0.923, n = 265) in the data obtained from unprocessed fresh octopus samples [19]; however, in the current study, we have also sampled frozen octopus (thawed before the experiments) to better represent the reality of octopus market.

### 3.4. Electrical Conductivity

Control octopus samples had the highest values of electrical conductivity (7.03 ± 1.36 mS·cm^−1^; Table 1), and in water-added octopus, the increase in immersion time caused a decrease in electrical conductivity until 3.31 ± 0.52 mS·cm^−1^ in the longest treatment (32 h). These changes reflected the gradual increase in water uptake and the decrease in ion concentration available in the interstitial fluid, as expected. A similar trend was reported in a previous study, although with higher absolute values in control octopus [19]. This might be explained by the loss of ions during thawing in octopus sampled in the current study.

Although significant differences were observed in the electrical conductivity of octopus, both control and water-added octopus shared a wide range of values, thus hampering the distinction between both groups based on this parameter.

The prediction of water uptake in octopus based on electrical conductivity was suggested previously, which could be an advantage to simplify the analysis [19]. However, in the current study, a weaker negative correlation was obtained (water uptake = −5.5061 × EC + 38,545; r = −0.53; *p* value < 0.01), possibly related to the different conditions of samples (frozen vs. fresh). Considering that the current study was performed with fresh and frozen octopus samples, to better represent the actual conditions of octopus available for consumers, the electrical conductivity per se as a method for quantitation of water uptake or moisture content is not suitable due to the higher dispersion of data obtained in comparison with the one reported by Mendes et al. [19].

### 3.5. Time Domain Reflectometry

#### 3.5.1. Data Exploration of TDR Results

In the case of destructive methods, control and water-added octopus samples shared a range of moisture (and protein) content values, whereas it would not be possible to discriminate if samples were or were not processed with water. However, the TDR data acquired with the Sequid RFQ-Scan® equipment show that control samples had a clearly different profile from water-added octopus, particularly in the region between 1.2 and 1.5 ns (Figure 3). In this region, the reflected TDR signal increased with the increase in immersion time, although some treatments showed comparable values (e.g., 0.5 and 1 h; 24 and 32 h), suggesting that even short time treatments can be discriminated using this non-destructive method. Changes due to salt content are mainly expected between 1 and 2.3 ns [28], and thus the differences observed seem to reflect the decrease in salt content as freshwater was incorporated in octopus. In general, the dielectric constant and dielectric loss factor of foods are increased with an increase in the water amount of foods, because water (the dipolar molecule) dominantly affects changes in the dielectric properties of food [20]. However, it is also known that as water is added, other constituents, notably ionic salts, become diluted and diffuse into the exterior water, causing a fall in dielectric loss factor at low frequencies [29]. The increase in water and salt contents in dry-cured hams results in a lower reflected signal at the end of the TDR curve [28]. This modification of the reflected signal is directly related to the non-linear increase in the loss factor, especially for frequencies below 1 GHz due to the ionic component of the Hasted–Debye model [28].

A multivariate analysis was performed with the TDR data of *Octopus vulgaris* samples in order to detect groups of octopus samples related to the processing treatments and with the moisture content of samples (Figure 4). PC1 and PC2 were plotted against each other for the different octopus samples, and 90.2% of the variance in the original data was explained by these two PCs together. Control samples were separated from water-added (on the basis of PC1). It was also observed that some separation among samples with the shortest (0.5–1 h) and longest (2–32 h) soaking times. The remaining PCs did not show useful clustering.

PC1 scores were tested to evaluate differences among processing time conditions by means of ANOVA, and specifically, the control was significantly different from all other treatments, and 0.5–1 h was significantly different from 2–32 h. The position of octopus samples in the plot showed a relation with moisture content, which was confirmed by the negative correlation found between PC1 and moisture content (r = −0.72; *p* value < 0.01). The variability obtained may be related with quality differences of frozen/thawed octopus samples between individuals and trials. Dielectric properties have been shown to have a direct relationship with post mortem time in muscle food, such as poultry meat, due to proteolytic processes [30]. Other studies also related to changes in microwave dielectric spectroscopy with quality parameters [31,32], and similar effects, could occur with octopus.

The results obtained with the TDR technology were corroborated by those of electrical conductivity, were discussed previously, and a significant positive correlation was determined between PC1 and electrical conductivity (r = 0.79; *p* value < 0.01). PC1 also showed a correlation with protein (r = 0.67; *p* value < 0.01) and M/P ratio (r = −0.64; *p* value < 0.01), although with a lower strength.

#### 3.5.2. Calibration and Validation of TDR Analysis

Several studies have reported different dielectric properties between control samples and water-added samples in different seafood products, including octopus [19] and scallops [33]. Dielectric properties were also able to discriminate natural from water-added samples in other muscle foods, such as chicken [33]. Moreover, modules of the TDR measuring device (Sequid RFQ-Scan®) are already available to detect if water was added to pangasius, Alaska pollock, redfish, turkey, and chicken [34]. In this sense, the TDR methodology was tentatively calibrated to distinguish between control and water-added *O. vulgaris*.

Research studies were also published dealing with the prediction of moisture content based on microwave dielectric spectroscopy for several food items, including alfalfa [35], onions [36], dry cured ham [28], chicken and scallops [33], among others. Taking into account that the reflected TDR data were correlated with the moisture content of octopus, a calibration of this methodology was further performed to allow a rapid and accurate quantitation of moisture content in *O. vulgaris*.

After calibrations, a new trial with *O. vulgaris* was conducted for validation purposes, with a total of 12 control and 36 water-added octopus samples. Water uptake, moisture and protein contents, and the electrical conductivity results are shown in Table 2. In this trial, octopus samples had moisture contents between 80.1 and 89.5 g/100 g. In particular, control samples had a moisture content of 81.6 ± 1.0 g/100 g, and it was not significantly different from the moisture content obtained in control octopus from the six calibration trials.

##### Detection of Water-Added Octopus (Control vs. Water-Added Octopus)

To discriminate control from water-added octopus, the principal components analysis was applied to the TDR data obtained in the calibration trials with *O. vulgaris*. The first two principal components explained 99.5% of the data’s variance, and two clusters were determined heuristically with an ellipsoid shape to include control octopus in one cluster, and water-added octopus in the other cluster. The two cluster parameters are shown in Table 3.

After calibration, new samples (from the validation trial) were represented in the principal components plot, and the probability of the samples belonging to the water-added cluster were estimated, taking into account the distance between clusters (Figure 5).

Analysis of the data shows that all control samples were correctly classified, as samples were positioned in the cluster of control octopus, and the maximum value of likelihood of being classified as water-added was 1.4%. Regarding water-added octopus, none of the samples were positioned in the cluster of control octopus, and as expected, higher likelihood values were obtained for these samples (Figure 5). Most water-added octopus samples (immersed for 2, 16, and 32 h) from the validation set showed reflected TDR signals similar to those of water-added octopus calibration samples, and thus were positioned in the cluster of water-added octopus. All octopus samples immersed for a longer period (32 h) were positioned in the water-added octopus cluster in the farthest area in relation to the control cluster, and these samples had the highest likelihood values (99.9 ± 0.1%). The results obtained showed the potential of the TDR technology to rapidly identify if *O. vulgaris* was processed with an addition of water and specially to discriminate octopus immersed in freshwater for longer periods.

##### Quantitation of Moisture Content of Octopus

To estimate the moisture content of octopus, a multiple linear regression was performed, using the moisture content of octopus samples obtained by the destructive method and the scores of the first two principal components obtained from the TDR data. The moisture content can be estimated by the following equation:Moisture content (g/100 g) = −0.503 × PC1 + 1.602 × PC2 + 85.35

The root mean squared error (RMSE) between the reference moisture content (obtained with the destructive method) and the moisture content predicted by the principal component regression calibration was 1.1%, and the coefficient of determination R^2^ was 0.784 (*p* < 0.05).

A group-wise internal cross validation was performed, and the error of validation was calculated using one trial for validation and the remaining ones for calibration, and then the procedure was repeated until all trials were used for validation (a total of six trials). The RMSE of the cross validation was 1.2% of moisture content.

The validation of the multiple linear regression calibration was also performed using another set of octopus samples (validation trial). Results obtained with the calibration were confronted with the values obtained with the classic destructive method (Figure 6), and the RMSE determined was 1.0% of moisture content.

Octopus samples from the validation trial were considered as hypothetical commercial samples, and moisture content results obtained with the TDR calibration were compared for conformity assessment with the proposed threshold value of 85.2 g/100 g, which limits the maximum amount of non-labeled added water (5%). The results show that all control octopus samples and most from the 2 and 16 h immersion treatments had values up to 85.2 g/100 g, including the uncertainty of the measurement (1.1%, RSME from the calibration estimation). Thus, the evaluation of the results obtained with the TDR methodology and comparison with the proposed threshold value of 85.2 g/100 g, allows one to conclude that in those cases, the label may omit if water has been added. Conversely, all samples from the 32 h treatment needed to have, in the label, the information that water was added.

In a previous study with octopus from the Portuguese market, 85% of samples had moisture contents higher than 86.0g/100 g [11]. It shows that the industry incorporates high amounts of water in octopus, and thus indicates that most commercial products would fall in the region near samples from the 32 h treatment.

Although the determination of the moisture content using TDR technology is associated with higher uncertainties than the classic destructive method, being rapid and non-destructive are important advantages. This would allow one to control, in a short period, a higher number of octopus products without damaging the products, and as such, lead industries to include the appropriate information on the label and/or to valorize the commercialization of products without added water.

## 4. Conclusions

In this study, a maximum limit of moisture content of 85.2 g/100 g in octopus products was proposed for conformity assessment, unless the label had the indication that water was added (more than 5%). This value was obtained using a total of 342 control *O. vulgaris* samples from the Portuguese coast, and it took into account the highest value of the 95% confidence interval of moisture content distribution data plus 5% added water.

The TDR methodology was calibrated with the purpose of detection and quantitation of water in *O. vulgaris*, thus showing that a rapid and non-destructive methodology based in changes in the dielectric properties of the muscle can be used for such an aim. Thus, microwave dielectric spectroscopy using TDR analysis can discriminate between control and water-added octopus, especially for octopus immersed in freshwater for longer periods (32 h). In addition, moisture content can be quantified in *O. vulgaris* with TDR analysis, in the range 80–90 g/100 g, with a RMSE of 1.1%. Each analysis with the TDR technology is performed within a few seconds, and it does not perforate or cause any other damage to octopus.

In the current study, different moisture contents were obtained by immersion of octopus only in freshwater. The use of additives (e.g., polyphosphates, salts), applied by the industry and known to affect the dielectric properties, should be tested in future research studies to determine their influence in the calibrations performed. Then, it would also be important to verify the adequacy of the calibrations performed with real commercial samples, which can be processed with different soaking treatments. TDR data of other cephalopods (e.g., other octopus species, cuttlefish), also suspected of water addition practices, used as substitute species and of commercial relevance, could also be investigated to determine the potential of this technology as a non-destructive analysis to determine if cephalopods were processed with added water/additives, to evaluate the need of more specific calibrations, and to verify compliance with legislation.

## Figures and Tables

**Figure 1 foods-11-00791-f001:**
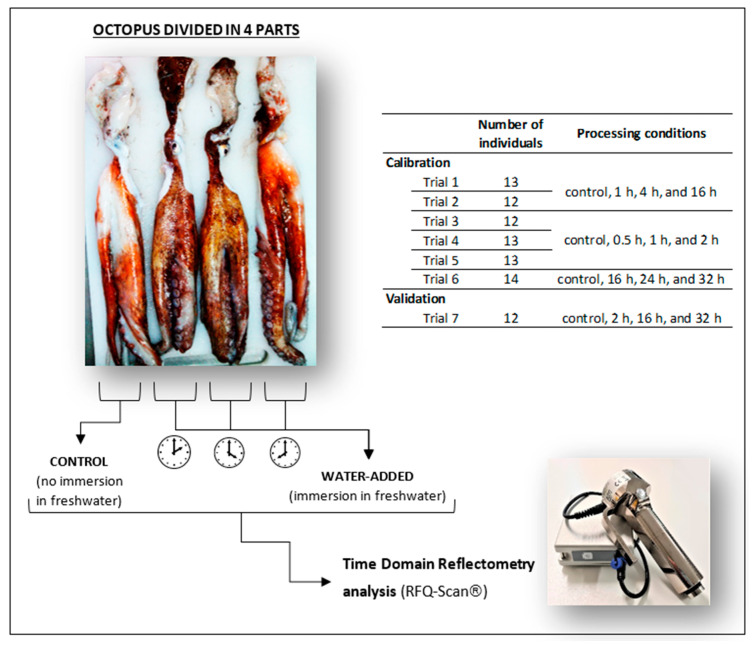
Schematic representation of water addition trials with *Octopus vulgaris*. Each octopus was divided into 4 parts, with one part used as the control, and the three remaining parts immersed for different periods (between 0.5 and 32 h) in freshwater. Time domain reflectometry analysis was performed with an RFQ-Scan® device. A total of six trials with 12–14 octopus were performed for the calibration of the device, and one trial with 12 octopus was performed for the validation.

**Figure 2 foods-11-00791-f002:**
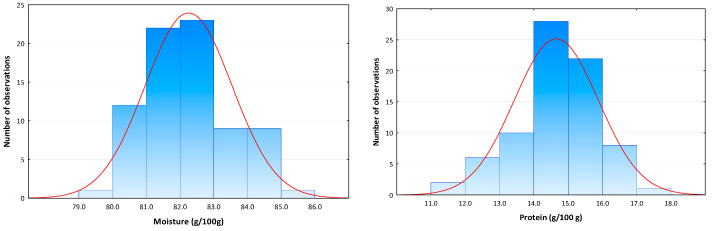
Histograms of moisture and protein content distribution in control *Octopus vulgaris* samples (n = 77) with fitted normal curve.

**Figure 3 foods-11-00791-f003:**
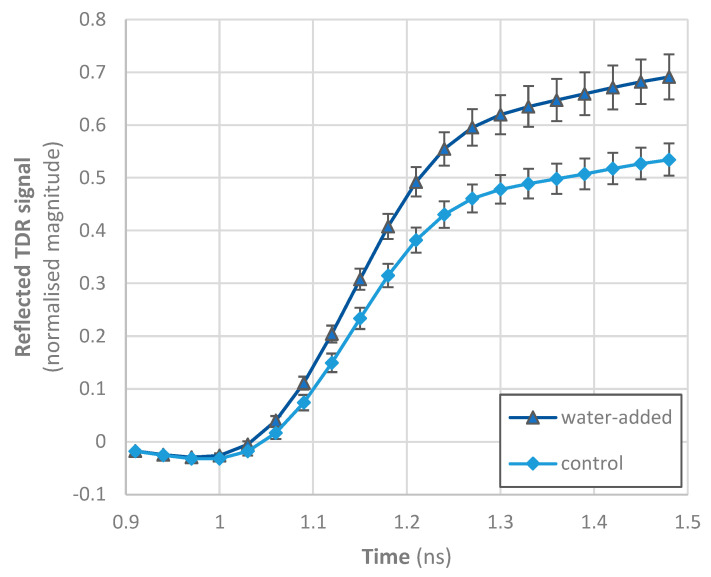
Time domain reflectometry data of *Octopus vulgaris* samples. Immersion time of water-added treatments varied from 0.5 to 32 h. Data represent the average and standard deviation of all samples of calibration trials.

**Figure 4 foods-11-00791-f004:**
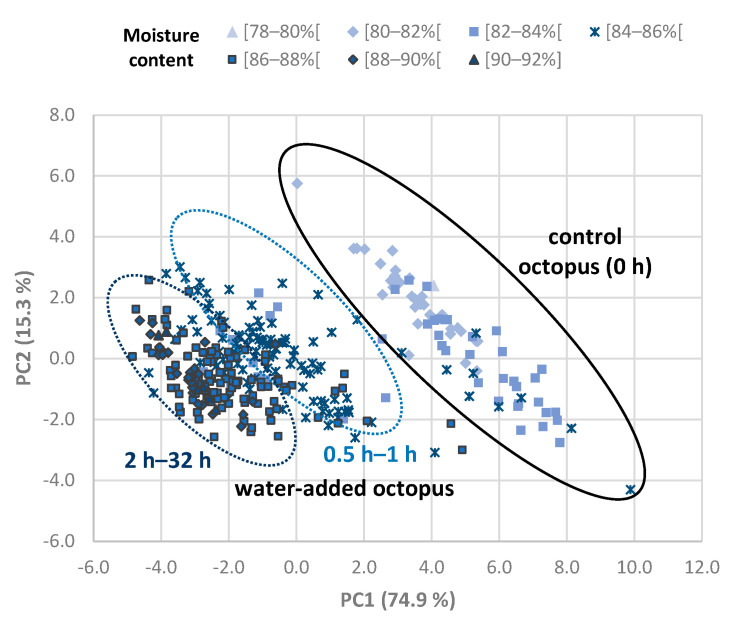
Principal components analysis of TDR data of *Octopus vulgaris* samples. Immersion time of water-added treatments varied from 0.5 to 32 h. Octopus samples from the calibration trials are represented in the plot as a function of moisture content (between 79.7 and 90.6 g/100 g).

**Figure 5 foods-11-00791-f005:**
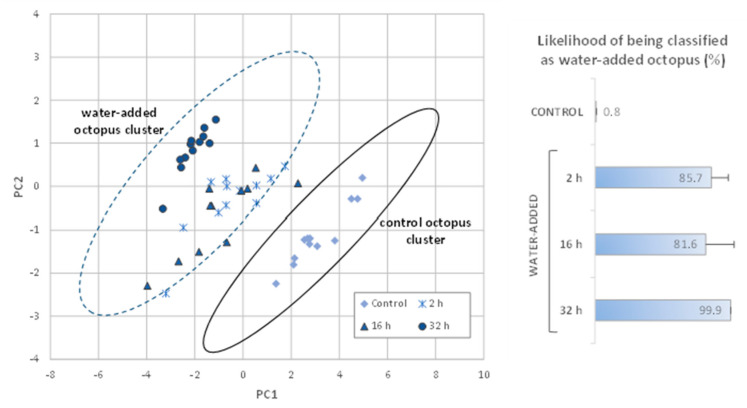
Validation results obtained with *Octopus vulgaris* samples represented in the plot of calibration clusters (control and water-added) and likelihood (mean ± SD) of being classified as water-added octopus. Calibration of TDR methodology was performed by means of principal components analysis—distance to reference. In the validation trial, water-added octopus were immersed in freshwater for 2, 16, and 32 h. PC1 and PC2 explained 99.5% of the variation of the original variables.

**Figure 6 foods-11-00791-f006:**
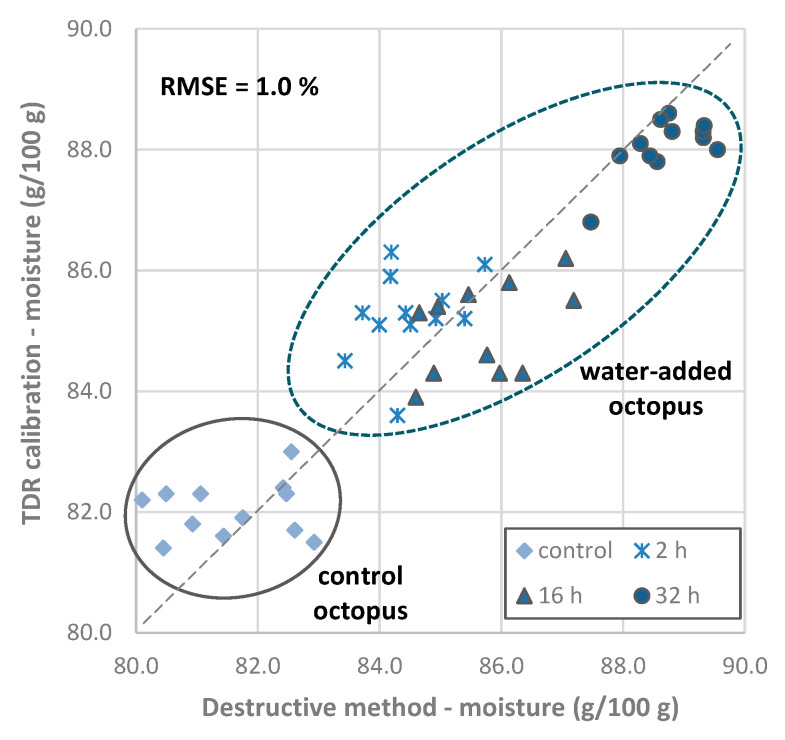
Validation results of *Octopus vulgaris* samples obtained using the TDR calibration for quantitation of moisture content vs. moisture content determined by the classic destructive method (predicted vs. actual values). The coefficient of determination R^2^ was 0.796. Abbreviations: RMSE, root mean squared error.

**Table 1 foods-11-00791-t001:** Characterization of *Octopus vulgaris* samples used in water addition trials for the calibration of TDR analysis. Water-added samples were immersed in freshwater for 0.5 to 32 h. Different superscript letters (A, B) denote significant differences between control and water-added octopus. Abbreviations: M/P ratio—moisture/protein ratio.

	Number of Samples	Moisture (g/100 g)	Protein (g/100 g)	M/P Ratio	Electrical Conductivity (mS·cm^−1^)
**Control Octopus**	77				
Mean ± SD		82.2 ± 1.3 ^B^	14.6 ± 1.2 ^A^	5.7 ± 0.6 ^B^	7.03 ± 1.36 ^A^
Min		79.7	11.0	4.7	4.72
Max		85.8	17.0	7.8	12.22
**Water-Added Octopus**	231				
Mean ± SD		86.3 ± 1.6 ^A^	11.7 ± 1.4 ^B^	7.6 ± 1.2 ^A^	4.67 ± 0.88 ^B^
Min		82.5	7.7	5.4	2.45
Max		90.6	15.3	11.8	7.60

**Table 2 foods-11-00791-t002:** Water uptake, moisture and protein contents, and electrical conductivity of *Octopus vulgaris* samples from the validation trial. Twelve octopus were sampled for control and for each immersion time treatment. Different superscript letters (A–D) denote significant differences between immersion time treatments. Abbreviations: M/P ratio, moisture/protein ratio.

	Immersion Time in Freshwater	Water Uptake (%)	Moisture (g/100 g)	Protein (g/100 g)	M/P Ratio	Electrical Conductivity (mS·cm^−1^)
**Control**	0 h	-	81.6 ± 1.0 ^D^	14.9 ± 0.8 ^A^	5.5 ± 0.4 ^C^	5.99 ± 1.15 ^A^
**Water-added**	2 h	7.5 ± 0.9 ^C^	84.5 ± 0.7 ^C^	12.3 ± 0.5 ^B^	6.9 ± 0.3 ^B^	4.74 ± 0.60 ^B^
	16 h	13.2 ± 2.9 ^B^	85.7 ± 0.9 ^B^	12.5 ± 1.2 ^B^	6.9 ± 0.7 ^B^	4.27 ± 0.49 ^B,C^
	32 h	31.3 ± 4.5 ^A^	88.7 ± 0.6 ^A^	10.3 ± 0.9 ^C^	8.7 ± 0.8 ^A^	3.83 ± 0.56 ^C^

**Table 3 foods-11-00791-t003:** Parameters of clusters determined for the calibration of TDR equipment to classify *Octopus vulgaris* as control or water added. PC1 and PC2 were the first two principal components obtained, and together explained 99.5% of the variation of the original variables.

	Cluster Center (PC1, PC2)	Radii (r_long_/r_short_)	Angle (rad)
Control octopus cluster	3.3, −1.0	5.0/1.0	0.503
Water-added octopus cluster	−1.8, 0.1	5.5/1.7	0.503

## Data Availability

Not applicable.

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
