# Peer review of "Quantitation of Water Addition in Octopus Using Time Domain Reflectometry (TDR): Development of a Rapid and Non-Destructive Food Analysis Method"

_foods, 2022, doi:10.3390/foods11060791_

Round 1

Reviewer 1 Report

The manuscript presents methodology for the quantification of water addition in octopus using time domain reflectometry (TDR). The proposed method is fast, non-destructive, and it enables the evaluation of water content in the octopus muscle to control the abusive addition of water and the compliance with regulations in this topic. The manuscript is well written, and the results are of interest. However, some concerns should be addressed.

  • Section 2.1 and Figure 1: Please, explain in a clearer way the number of trials, individuals and parts, controls and validations. It is hard to understand how they were distributed in the different test conditions.
  • Section 2.2 and 2.3: Used methods should be explained further. Please, give full experimental procedures and details.
  • Is it possible to use % units instead of g/100g?

Author Response

We wish to express our gratitude to the reviewers for their insightful comments and thorough review, which have helped us to improve the paper.
In the attached files, replies to reviewers’ comments are addressed point–by-point, and in the revised manuscript the changes were made using "track changes".
Thank you for your kind attention.
Best regards

Reviewer 2 Report

This is well, clearly and neatly written paper. The topic is actual and interesting in the context of ensuring of fair producers' practices and consumer's interest. The authors justified the research undertaken, the experiment was well designed, the results were documented, presented and analysed correctly.

Please correct the: 0.52 mS.cm-1 (page 8).
In my opinion this work is fully justified.

Author Response

(The authors gave the same response as above.)

Reviewer 3 Report

The authors investigated a rapid and non-destructive method based in Time Domain Reflectometry analysis (TDR),that detects and quantifies the water content in the muscle, was developed for the control of abusive water addition to octopus. The authors obtained interesting/promising results . Their work can further improve by:

a) Introduction could provide additional literature on what brought about TDR a strong non-destructive detection method, why is the commercial status of octopus worthy of investigation from the context of consumer demand, more justification as to why this study is relevant?

b) Materials and methods is ok. Please in the results and discussion, there is a need for more  discussion using relevant synthesized  literature . Please, authors kindly try to provide more answer to the questions of why and how? and not only where and what, in the discussion of results ...don’t just state literature but try to further contextualise it to the argument/debate

c) Conclusions should please provide direction for future studies, based on the challenges authors might have countered in the course of undertaking this work.

Look forward to your revised manuscript ?

Author Response

(The authors gave the same response as above.)
